# Molecular Mechanisms of Hyperoxia-Induced Neonatal Intestinal Injury

**DOI:** 10.3390/ijms24054366

**Published:** 2023-02-22

**Authors:** Hsiao-Chin Wang, Hsiu-Chu Chou, Chung-Ming Chen

**Affiliations:** 1Department of Pediatrics, Shuang Ho Hospital, Taipei Medical University, New Taipei 235, Taiwan; 2Department of Pediatrics, School of Medicine, College of Medicine, Taipei Medical University, Taipei 110, Taiwan; 3Department of Anatomy and Cell Biology, School of Medicine, College of Medicine, Taipei Medical University, Taipei 110, Taiwan; 4Department of Pediatrics, Taipei Medical University Hospital, Taipei 110, Taiwan

**Keywords:** hyperoxia, intestinal injury, reactive oxygen species, nitric oxide, cytokines

## Abstract

Oxygen therapy is important for newborns. However, hyperoxia can cause intestinal inflammation and injury. Hyperoxia-induced oxidative stress is mediated by multiple molecular factors and leads to intestinal damage. Histological changes include ileal mucosal thickness, intestinal barrier damage, and fewer Paneth cells, goblet cells, and villi, effects which decrease the protection from pathogens and increase the risk of necrotizing enterocolitis (NEC). It also causes vascular changes with microbiota influence. Hyperoxia-induced intestinal injuries are influenced by several molecular factors, including excessive nitric oxide, the nuclear factor-κB (NF-κB) pathway, reactive oxygen species, toll-like receptor-4, CXC motif ligand-1, and interleukin-6. Nuclear factor erythroid 2-related factor 2 (Nrf2) pathways and some antioxidant cytokines or molecules including interleukin-17D, n-acetylcysteine, arginyl-glutamine, deoxyribonucleic acid, cathelicidin, and health microbiota play a role in preventing cell apoptosis and tissue inflammation from oxidative stress. NF-κB and Nrf2 pathways are essential to maintain the balance of oxidative stress and antioxidants and prevent cell apoptosis and tissue inflammation. Intestinal inflammation can lead to intestinal damage and death of the intestinal tissue, such as in NEC. This review focuses on histologic changes and molecular pathways of hyperoxia-induced intestinal injuries to establish a framework for potential interventions.

## 1. Introduction

Supplemental oxygen is often used to treat respiratory disorders in newborns; however, high concentrations of oxygen can have both beneficial and adverse effects. In rabbits, liver, and kidney tissue oxygenation increased due to inhalation of a 100% oxygen mixture [1]. High tissue oxygenation increases oxidative stress and leads to tissue injury [2]. Prolonged exposure to normobaric hyperoxia for 2 to 4 days induced lung, kidney, and ileum injury in newborn rats [3]. Previous studies in pediatric medicine have focused on the mechanisms underlying hyperoxia-induced lung injury [3,4,5,6]. In the neonatal intensive care unit (NICU), food intolerance and necrotizing enterocolitis (NEC) are the important problems. The early postnatal stress, physiological inflammatory responses, and microbiota alterations induced by infections or early antimicrobial use all caused impaired development of the gastrointestinal tract [7]. Healthy gut development and microbiome in neonate play indispensable roles in development of brain and immune systems [8,9,10]. This paper reviews the effects of hyperoxia on intestinal development and the mechanisms mediating these effects in newborn animals. A thorough understanding of the molecular mechanisms underlying hyperoxia-induced intestinal injury will help identify novel therapeutic targets in hyperoxia-associated intestinal injury.

## 2. Embryonic Intestine Development

The primitive gut tube develops through the incorporation of the yolk sac during craniocaudal and lateral folding of the embryo [11,12]. This tube is then divided into three distinct sections: foregut, midgut, and hindgut. The foregut gives rise to the esophagus, stomach, liver, gallbladder, bile ducts, pancreas, and proximal duodenum; the midgut gives rise to the distal duodenum, jejunum, ileum, cecum, appendix, ascending colon, and proximal 2/3 of the transverse colon; and the hindgut gives rise to the distal 1/3 of the transverse colon, descending colon, sigmoid colon, and upper anal canal. The intestinal epithelium, which is the largest surface area of the body, is the major crucial barrier against the outside environment. The intestinal epithelium forms a barrier between the intestinal lumen and the interstitium [13]. Humans have long gestation periods; the gastrointestinal tract is mostly formed during gestation [7,14,15].

In rodents that have short gestation periods, such as rats and mice, the intestine is relatively immature at birth and it becomes mature 2 weeks after birth [16,17]. These features increase the susceptibility of newborn rodents to hyperoxia. Rodents are suitable animal models for studying O_2_ toxicity-related acute intestinal damage.

## 3. Intestinal Histological Changes after Hyperoxia

In a mouse model, maternal inflammation during pregnancy altered the gastrointestinal tract of offspring [18]. The gastrointestinal tract plays a key role in innate immunity [19,20]. Intestinal villi and mucosae continue to grow and differentiate after birth [21]. Previous studies of hyperoxia-induced intestinal injury were focused on small intestines and large intestines. Hyperoxia increased ileal mucosal thickness ad induced separation of lamina propria from submucosa [22,23], and fewer Paneth cells, goblet cells, and villi in the intestinal epithelium were observed [18,24,25,26,27]. In neonatal rats reared in an O_2_-enriched environment, enterocytes were shorter, and the surface of apical cells was flattened [28]. Intestinal secretory components decrease under conditions of hyperoxia. Intestinal secretory components and proteins play vital roles in the mucosal immune system. Secretory components increase the viscosity of mucus, which facilitates mucosal adhesion and the mucosal immunological defense system, thereby preventing pathogen adhesion to host cells and limiting inflammation [29,30]. Secretory components also prevent proteolytic degradation of secretory immunoglobulin A (SIgA), which weakens bacterial translation [31]. Polymeric immunoglobulin receptor (pIgR) transfers soluble dimeric IgA, pentameric IgM, and immune complexes from the basolateral to the apical mucosal epithelial cell surface [32]. PIgR plays a crucial role in intestinal defense against pathogenic microbes [33]. Moderate oxygen induced an increase and hyperoxia induced intestinal secretory components in neonatal rats and this might be brought to increase the intestinal SIgA. Large amounts of secretory components and SIgA may help in maintaining optimal conditions against pathogens [22].

The degrees of vessel dilation, vascularization, inflammation, and fibrosis were significantly increased under conditions of hyperoxia [3]. Misalignment and distension of the basolateral intercellular space in the neonatal rat’s epithelium were also noted [22], and epithelial cell apoptosis and the mortality rate significantly increased [32,34]. The small intestine in neonates is sensitive to excess oxygen. In a rat study, neonatal hyperoxia exposure resulted in injury to the small intestine and disruption of the intestinal barrier during the first 2 postnatal weeks [35]. The densities of Zonula occludens (ZO)-1, occludin, and claudin-4/β-actin were lower in the hyperoxia condition [36], indicating intestinal tight junction dysfunction. Moreover, serum interleukin (IL)-6 levels increased [37], indicating inflammation. This may influence the intestinal barrier and reduce the clearance of bacterial pathogens [18,38]. Paneth cells produce cathelicidins, which are antimicrobial peptides that protect the tight junction from hyperoxia. The number of Paneth cells also decreased in the neonatal rat [22]. Damage to the intestinal barrier leads to the absorption of lipopolysaccharides (LPS) and aggravates bacterial invasion. ZO-1, occludin, cingulin and claudin-4 were significantly down regulated in NEC preterm neonates [24]. Intestinal barrier dysfunction is a major predisposing factor in the development of NEC [28,39,40,41,42].

## 4. Mechanisms Associated with Hyperoxia-Induced Intestinal Injury (Table 1)

### 4.1. Nitric Oxide (Figure 1)

Nitric oxide (NO) is a free radical and can be a proinflammatory mediator as a signaling molecule, interacting with oxygen to became oxidized. This reaction is regulated by nitric oxide synthases (NOS), inhibits platelet adhesion, prevents mast cell activation, and functions as an antioxidant [43,44]. High NO levels disrupt actin cytoskeletons, inhibit ATP formation, dilate cellular tight junction, and increase intestinal permeability [45]. Under hyperoxic conditions, NO dysregulation occurs and NOS II protein concentrations in the villus, crypts, submucosa, and muscularis are increased [23]. Excessive NO production leads to mucosal injury and may cause NEC [43,46,47].

### 4.2. Nuclear Factor-κB (Figure 1)

Nuclear factor-κB (NF-κB) is a protein complex that controls DNA transcription and regulates the expression of cytokines, inducible NOS, and cyclooxygenase (COX)-2 [48]. NF-κB is activated in response to many external stimuli, including cytokines, free radicals, and bacterial or viral antigens. NF-κB is a transcription factor and plays a crucial role in immune response and inflammation [48,49]. Under hyperoxic conditions, NF-κB transcription increases, which induces the production of proinflammatory mediators, such as tumor necrosis factor (TNF)-α and interferon (IFN)-γ [30,50,51]. Excessive NF-κB expression may be an important inflammatory mechanism of NEC [52] and may be related to inflammatory bowel disease [45,53]. NF-κB can directly repress Nrf2 signaling (the protective mechanism, it will be discussed in part 5.1) to increase inflammation [54].

### 4.3. Reactive Oxygen Species (Figure 1)

Reactive oxygen species (ROS) are a chemically defined group of reactive molecules derived from molecular oxygen and ROS are produced by mitochondria. Low levels of ROS are related to cell proliferation and differentiation. Hyperoxia is associated with increased production of ROS, and excessive ROS induces apoptosis, cell autophagy, and DNA oxidative damage. A redox imbalance under hyperoxic conditions causes inflammation via the NF-κB and TNF pathways [32,44], which leads to an intestinal inflammation cascade and, ultimately, mucosal damage [36]. Mucosal damage induces the production of proinflammatory cytokines including IFN-g and IL-1. ROS also can modify cellular structure and function through covalent changes of NO and cause NO dysregulation [55]. Under conditions of hyperoxia, ROS promotes inflammatory cascades in the gut and may relate to inflammatory bowel disease [24,56].

### 4.4. Toll-like Receptor (Figure 1)

Toll-like receptors (TLRs) are membrane-spanning proteins and TLR4 is activated by bacterial LPS. TLRs are pathogen recognition receptors and play a critical role in the early innate immune response to invading pathogens [57,58]. Thus, they initiate innate immune and inflammatory responses. TLR4 also mediate NEC [40,59]. Hyperoxia increases the expression of TLR4 and NF-κB, thereby inducing inflammatory reactions and leading to intestinal injury via the NF-κB pathway [52]. The ROS levels also increase. A cascade of proinflammatory cytokines and interferons then begins, which induces inflammation [28]. Gut-derived endotoxemia may contribute to lung injury via the TLR4 pathway under conditions of hyperoxia [24]. Inhibition of the TLR pathway may prevent NEC [60].

### 4.5. Chemokine (CXC Motif) Ligand (Figure 1)

CXC motif ligand (CXCL) 1 is a member of the CXC chemokine family. It plays a vital role in the development of many inflammatory diseases [61]. It activates the CXC receptors (CXCR) 1 and 2. CXCL1 transcription involves the NF-κB pathway and cytokines such as IFN-γ, IL-1β, IL-17, transforming growth factor-β, and TNF-α [62,63].

The level of CXCL1 increases under inflammatory conditions and induces angiogenesis and the recruitment of neutrophils. Gut neutrophilia may induce gut injury under conditions of hyperoxia [24].

### 4.6. IL-6 (Figure 1)

IL-6 is a potent cytokine that modulates the innate immune system. Fetal exposure to maternal inflammation significantly increases the susceptibility and severity of subsequent intestinal injury with goblet cell loss via IL-6 [18,24]. Hyperoxia increases IL-6 levels in intestinal epithelial cells and aggravates inflammatory responses [47]. The hallmark of vascular NF-κB activation is the production of IL-6, which may play a role in vascular inflammation [64]. IL-6 is associated with intestinal barrier dysfunction and intestinal injury [37].

**Table 1 ijms-24-04366-t001:** Evidence of the roles of various molecules in the intestine under hyperoxic conditions.

**Injury Mechanism**				
**Candidate**	**Mechanism**	**Model**	**Outcome**	**Reference**
Excessive Nitric oxide (NO)	Signaling molecule Proinflammatory mediator	Rat; Piglet	Disrupts actin cytoskeletons Inhibits ATP formation Dilates tight junction May cause NEC	[23,45,47]
Nuclear factor-κB (NF-κB)	Protein complex Transcription factor	Human HT-29 cell; Rat	Regulates the expression of NOS and COX2 cytokines Produces proinflammatory mediators, such as TNF-α and IFN-γ Repress Nrf2 signaling May be related to NEC and IBD	[30,52]
Reactive oxygen species (ROS)	Signaling molecule Proinflammatory mediator	Rat; Human Caco-2 cell; Human NCM460 cell	Induces apoptosis, cell autophagy, and DNA oxidative damage Increases intestinal inflammation and mucosal damage Increases levels of TNF-α and IL-1 May be related to IBD	[24,32,36,56]
Toll-like receptor-4 (TLR4)	Transmembrane protein Trigger of signaling cascades of proinflammatory cytokines	Rat; Human cell	Initiates innate immune response and inflammatory reaction Increases ROS level	[40,52,59]
CXC motif ligand-1 (CXCL1)	Member of the CXC chemokine family	Rat	Transcription with NF-κB pathway and inflammatory cytokines Induces angiogenesis and recruitment of neutrophils	[24]
Interleukin-6 (IL-6)	Cytokine that modulates the innate immune system	Rat; Piglet	Aggravates inflammatory response Causes intestinal injury with goblet cell loss	[18,24,47,64]
**Protective Mechanism**				
**Candidate**	**Mechanism**	**Model**	**Outcome**	**Reference**
Nuclear factor erythroid 2-related factor 2 (Nrf2)	Transcription factor	Rat; Piglet; Human NCM460 cell; Human F244 cell; Murine embryonic fibroblasts cell	Regulates the expression of multiple antioxidant genes ROS detoxification and scavenging Decreases cellular oxidative damage Maintains the balance of intracellular redox status Interferes with IL-6 induction Promotes IL-17D production	[54,65,66]
Interleukin-17D (IL-17D)	Cytokine that modulates inflammation and host defense Anti-inflammatory cytokine	Rat; Human F244 cell; Murine embryonic fibroblasts cell	Induces CXCL2 expression Causes recruitment of NK cells Inhibits the bacterial phagocytic ability of macrophages Inhibits and regulate TNF-α and other proinflammatory cytokines Delays the intestinal inflammatory response and protects the GI tract from damage	[65,66]
N-acetylcysteine (NAC)	Powerful antioxidant Scavenger of hydroxyl radicals	Rat	Anti-inflammatory activities Inhibits ROS Reduces hyperoxia-induced ZO-1, occludin, and claudin-4 damage Improves intestinal barrier	[36]
Arginyl-glutamine Arg-Gln	Dipeptide	Rat; preterm neonate	Preserves the actin cytoskeleton to maintain the function of the intestinal barrier and intercellular junction Anti-inflammatory Improves mucosal integrity and gut healing Reduces the risk of NEC	[67,68]
Deoxyribonucleic acid (DHA)	Omega-3 long-chain fatty acid	Rat	Reduces inflammation Blocks platelet-activating factor-induced apoptosis in intestinal epithelial cells Reduces the risk of NEC	[67,69]
Cathelicidin	Antimicrobial peptide	Rat; Piglet; Porcine epithelial cell line J2	Antibacterial, antiviral, and antifungal Inhibits hyperoxia-induced NF-κB pathway reaction Enhances the phagocytosis of immune cells Suppresses intestinal inflammation Reduces LPS-induced disruption of intestinal barrier	[22,70,71]
Health gut microbiome	Community of microorganisms in the gut Regulates the immune function and immune homeostasis	Rat; Human infant	Modulates NF-κB pathway reaction Prevents bacterial infection Reduces the risk of NEC Reduces lung inflammation May be related to neurodevelopmental disorders, neurodegenerative disorders, and metabolic syndromes	[24,38,72,73,74,75,76]

Abbreviations: ATP: adenosine triphosphate; COX2: cyclooxygenase 2; DNA: deoxyribonucleic acid; GI: gastrointestinal; IBD: inflammatory bowel disease; IFN-γ: interferon-γ; LPS: lipopolysaccharide; NAC: N-acetylcysteine; NEC: necrotizing enterocolitis; NK cell: natural killer cell; NOS: nitric oxide synthases; TNF-α: tumor necrosis factor-α; ZO-1: zonula occludens-1.

**Figure 1 ijms-24-04366-f001:**
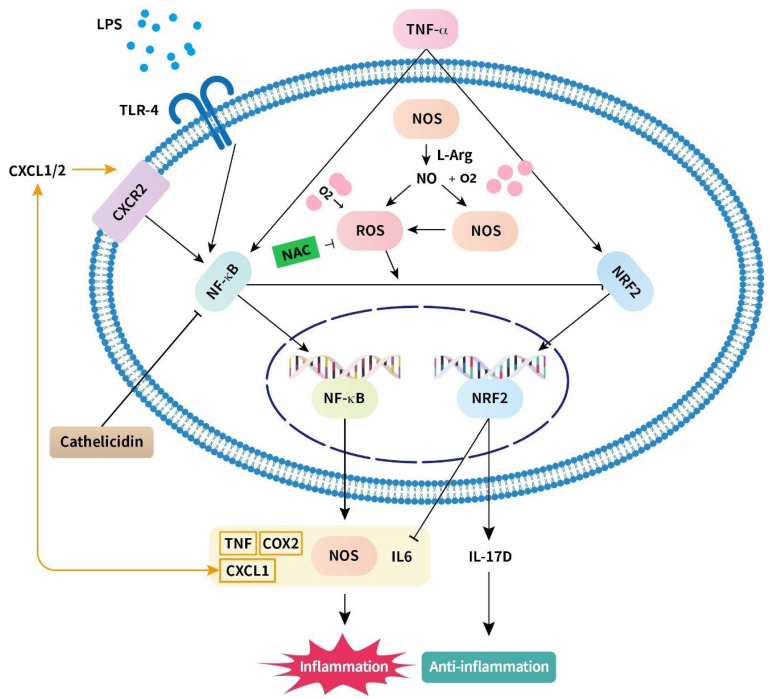
Schematic of the mechanism of hyperoxia-induced intestinal inflammation and anti-inflammatory response. Hyperoxia increases NOS and causes NO dysregulation and NO interacts with oxygen to became oxidated. Excessive NO induces the production of ROS. Furthermore, hyperoxia increases production of ROS. ROS causes inflammation through the NF-κB pathway and inhibits the Nrf2 pathway, which plays a crucial role in anti-inflammatory response. Production of inflammatory cytokines, including TNF, COX2, CXCL1, and IL-6, enhances inflammatory injury. CXCL1 activates CXCR1/2 and induces the NF-κB pathway. Under hyperoxic conditions, levels of TLR4 increase and bacterial LPS signal to enhance inflammation via the NF-κB pathway. Nrf2 inhibits the production of IL-6 and promotes the production of IL-17D, which may play a role in the production of anti-inflammatory responses. L-Arg inhibits apoptosis via increased NO production and antioxidant capacity and via inhibition of inflammation mediated by the NF-κB pathway. IL-10 inhibits TNF-α, and other proinflammatory cytokines delay intestinal inflammatory responses. NAC inhibits the ROS system to protect intestinal epithelial cells from hyperoxia. Abbreviations: Arg: arginine; COX2: cyclooxygenase 2; CXCL-1: chemokine (C-X-C motif) ligand-1; CXCR: C-X-C motif chemokine receptor; IL: interleukin; LPS: lipopolysaccharide; NAC: N-acetylcysteine; NF-κB: nuclear factor-κB; NO: nitric oxide; NOS: nitric oxide synthases; Nrf2: nuclear factor erythroid 2-related factor 2; ROS: reactive oxygen species; TLR-4: toll-like receptor-4; TNF: tumor necrosis factor.

## 5. Protective Mechanism under Hyperoxic Conditions (Table 1)

### 5.1. Nuclear Factor Erythroid 2-Related Factor 2 and IL-17D (Figure 1)

Nuclear factor erythroid 2-related factor 2 (Nrf2) initiates the transcription of downstream regulatory antioxidant proteins and regulates the expression of multiple antioxidant genes [77]. It is involved in ROS detoxification and scavenging, reduces cellular oxidative damage, and maintains the balance of intracellular redox [65].

IL-17D is a cytokine of the IL-17 family and is involved in inflammation and the host defense system. IL-17D is produced by intestinal lymphocytes and epithelial cells of the small intestinal villi. It induces CXCL2 expression and recruits natural killer cells, thereby initiating an antitumor or antiviral immune response. IL-17D directly inhibits the bacterial phagocytic ability of macrophages and aggravates sepsis via the NF-κB pathway [78]. IL-17D is a critical cytokine during intracellular bacterial and viral infection [65,79].

Nrf2 promotes IL-17D expression; hyperoxia promotes the nuclear translocation of Nrf2 and increases Nrf2 with IL-17D in intestinal epithelial cells [66]. Nrf2 also interferes with IL6 induction and inflammatory phenotypes in vivo [80]. Nrf2 and NF-κB are key pathways regulating the balance of cellular redox status and responses to stress and inflammation. NF-κB can directly repress Nrf2 signaling to increase inflammation [54].

### 5.2. N-Acetylcysteine (Figure 1)

N-acetylcysteine (NAC) is a powerful antioxidant and a scavenger of hydroxyl radicals. NAC has anti-inflammatory properties. NAC protects intestinal epithelial cells under conditions of hyperoxia through the inhibition of ROS. Moreover, NAC can reverse the decreases in ZO-1, occludin, and claudin-4 induced by hyperoxia, indicating that NAC can improve the intestinal barrier status [36].

### 5.3. Arginyl–Glutamine (Figure 1)

Arginyl–glutamine (Arg–Gln) dipeptide is an aqueous stable source of glutamine. Glutamine (Gln) maintains the function of the intestinal barrier and intercellular junction by preserving the actin cytoskeleton. It also reduces the levels of proinflammatory cytokines and prevents cytokine-related apoptosis of intestinal epithelial cells, thereby reducing the risk of developing NEC in premature animals [78]. Arginine (Arg) has anti-inflammatory properties and improves mucosal integrity and gut healing. L-Arg inhibits apoptosis via increased NO production and antioxidant capacity and inhibition of inflammation mediated by the NF-κB pathway [81]. In hyperoxia-induced intestinal injury neonatal mice, the Arg-Gln supplementation group experienced less intestinal injury than the room air group [67]. Low Arg levels are associated with NEC, and therefore, Arg supplementation may reduce the risk of NEC in premature animals and neonates [67,68,82].

### 5.4. Docosahexaenoic Acid (Figure 1)

Omega-3 long-chain fatty acids such as docosahexaenoic acid (DHA) decrease inflammation. DHA blocks platelet-activating factor-induced apoptosis in intestinal epithelial cells [67]. Insufficient levels of DHA may predispose neonates to acute inflammatory conditions [83]. In hyperoxia-induced intestinal injury neonatal mice, the DHA supplementation group had less intestinal injury as the room air group [67]. Thus, DHA supplementation may reduce the incidence of NEC in neonate rats [69].

### 5.5. Cathelicidin (Figure 1)

Cathelicidin is an antimicrobial peptide with antibacterial, antiviral, and antifungal properties. It acts as a multifunctional effector molecule in innate immunity [84]. Cathelicidin enhances hyperoxia-induced phagocytosis in weaning piglets through inhibition of the hyperoxia-induced NF-κB pathway [70], suppresses intestinal inflammation, and prevents intestinal barrier dysfunction. It also reduces LPS-induced disruption of the intestinal barrier in rats [70,71]. Therefore, cathelicidin treatment can ameliorate intestinal injury by protecting the tight junction from hyperoxia [22].

### 5.6. Health Gut Microbiome

Gut microbiomes contribute to gut mucosal development and integrity through innate immunity. Microbiota play a crucial role in human health and disease. [85,86]. Gut microbiota and the microbial metabolites may affect immune function and immune homeostasis. Microbiota may be associated with NEC [87]. The composition of gut microbiota under conditions of hyperoxia may influence the levels of lung cytokines and is associated with lung inflammation [46]. Interactions between the brain and gut microbiota may also be associated with neurodevelopmental disorders, neurodegenerative disorders, and metabolic syndromes [72].

When mice were exposed to hyperoxia in the first week of life, their gut microbial composition changed. Alpha diversity and taxa of microbiota decreased until adolescence, and *Proteobacteria* abundance increased [38]. Hyperoxia-induced gut injury was time- and dose-dependent in a mouse model. Hyperoxia not only diminished obligate anaerobes but also enriched facultative anaerobes, such as *Escherichia-Shigella, Enterobacteriaceae, Gammaproteobacteria,* and *Proteobacteria* [24]. Moreover, in a human study, maternal chorioamnionitis was demonstrated to increase the incidence of late-onset sepsis and death among preterm infants and shifted the fecal microbiome of preterm infants [73]. Prenatal LPS exposure also induced significant changes in the intestinal microbiome of the offspring, with a significant increase in the abundance of *Proteobacteria (Escherichia-Shigella)* and a decrease in *Firmicutes* at 7 days after birth [74].

The microbiome is a promising target for prevention and treatment with probiotics. *Saccharomyces boulardii* was shown to modulate NEC in neonatal mice via the NF-κB pathway [75]. The incidence of NEC was decreased in preterm infants given probiotics [76].

## 6. Conclusions

In this study, we reviewed the mechanism underlying neonatal intestinal damage caused by hyperoxia at a histological and molecular level. Under conditions of hyperoxia, protective mechanisms are activated to prevent intestinal inflammation and injury. The NF-κB and Nrf2 pathways play key roles in intestinal inflammation. Under conditions of hyperoxia, the redox status becomes imbalanced, resulting in a cascade of intestinal damage. We discussed the importance of the microbiome in hyperoxia-induced intestinal damage. Supplemental oxygen therapy is essential for preterm neonates with respiratory distress syndrome and other pulmonary conditions. The optimal target range of oxygen saturation is 91–95% in infants less than 28 weeks gestation [88]. There was increased mortality and NEC in the low oxygen saturation target group (85–89%) compared to the high SpO_2_ group (91–95%) [89]. However, high concentrations of oxygen can have adverse effects on newborns. In conclusion, oxygen therapy is important for newborns and hyperoxia can cause intestinal histological changes including mucosal damage and vascular changes. Thus, caution in its use is required. Further investigation is warranted to clarify the effects of hyperoxia on neonatal intestines and to identify methods of reducing hyperoxia-induced intestinal damage.

## Data Availability

The data presented in this study are available on request from the corresponding author.

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
