# Peer review of "Molecular Mechanisms of Hyperoxia-Induced Neonatal Intestinal Injury"

_ijms, 2023, doi:10.3390/ijms24054366_

Round 1

Reviewer 1 Report

The paper by Dr. Hsiao-Chin Wang reviewed what kind of mechanisms were related in hyperoxia-Induced Neonatal Intestinal Injury. It contributes understanding of oxygen therapy. However the authors have description need to be clarified.

1.     It is unclear where table 1 and figure 1 were described in text. Please insert the words in text.

2.     Please clarify “SIgA and IgA may change” in line 69. What does the change mean? Ans also “SIgA were increased” in line 70 was described. It is unclear what the authors explain about SIgA.

3.     LPS can activate only TLR4. Please correct line 130.

4.     Reference 44 and 50 in line 168 were incorrect. Please add appropriate reference.

5.     Section 5.2 IL-10 describe just hypothesis. Please remove it.

6.     Please add description of all reference listed in table 1 in text.

Author Response

Author Response to Reviewer 1 Comments:

The paper by Dr. Hsiao-Chin Wang reviewed what kind of mechanisms were related in hyperoxia-Induced Neonatal Intestinal Injury. It contributes understanding of oxygen therapy. However the authors have description need to be clarified.

Response: Thanks for the reviewer’s comments.

  1. It is unclear where table 1 and figure 1 were described in text. Please insert the words in text.

Response: Parts 4 and 5 are mentioned the mechanism of injury and protection under hyperoxic condition. Based on the knowledge and reference from parts 4 and 5, we make Table 1 and Figure 1. We had marked them behind parts 4 and 5. Thanks for the reviewer’s advice.

  1. Please clarify “SIgA and IgA may change” in line 69. What does the change mean? Ans also “SIgA were increased” in line 70 was described. It is unclear what the authors explain about SIgA.

Response: We deleted “Under conditions of hyperoxia, intestinal SIgA and IgA may change. High level Under conditions of hyperoxia, SIgA and SC were markedly increased in neonatal rats” and added “Moderate oxygen induced an increase and hyperoxia induced intestinal secretory components in neonatal rats and this might be brought to increase the intestinal SIgA. Large amounts of secretory components and SIgA may help in keeping the optimal condition from pathogen [22].” on lines 71-74 in the revised manuscript.

  1. LPS can activate only TLR4. Please correct line 130.

Response: We corrected “Toll-like receptors (TLRs) are membrane-spanning proteins that are activated by bacterial LPS.” to “Toll-like receptors (TLRs) are membrane-spanning proteins and TLR4 is activated by bacterial LPS.” on lines 127-128 in the revised manuscript.

  1. Reference 44 and 50 in line 168 were incorrect. Please add appropriate reference.

Response: Thanks for the reviewer’s comments. Correct references were added in “Nrf2 promotes IL-17D expression; hyperoxia promotes the nuclear translocation of Nrf2 and increases Nrf2 with IL-17D in intestinal epithelial cells [39, 44, 59].” on line 180.

  1. Section 5.2 IL-10 describe just hypothesis. Please remove it.

Response: Thanks for the reviewer’s comment. We removed this section and remodified the figure in the revised manuscript.

  1. Please add description of all reference listed in table 1 in text.

Response: Thanks for the reviewer’s suggestion. We added “Nitric oxide (NO) is a free radical and can be a proinflammatory mediator as a signaling molecule to interact with oxygen to became oxidized,” on lines 95-96, and “NF-κB can directly repress Nrf2 signaling (the protective mechanism, it will be discussed in part 5.1) to increase inflammation [45].” on lines 112-113. We also added a new sentence from reference 62 “In hyperoxia-induced intestinal injury neonatal mice, Arg-Gln supplementation group had a lesser intestinal injury as room air group [62].” on lines 199-200 and “In hyperoxia-induced intestinal injury neonatal mice, DHA supplementation group had a lesser intestinal injury as room air group [62].” on lines 207-208. We also corrected the order of references.

Reviewer 2 Report

In this manuscript, Wang and colleagues review molecular mechanisms of hyperoxic injury to the neonatal intestine, which is particularly relevant to the clinical problem of necrotizing enterocolitis (NEC), which is a significant issue in premature newborns.

 The paper is generally well written, and it introduces the topic in a manner that is useful for both clinician scientists and bench researchers with an interest in hyperoxic injury and/or NEC. Below are some comments for the authors to consider, which may make the paper more accessible or useful to potential readers.

Major issues:

The major weakness of this paper is that while it addresses potential mechanisms of hyperoxic injury to the neonatal intestine that may contribute to NEC, it fails to address the clinical evidence on the association between intestinal oxygenation and intestinal injury.  Evidence from randomized, controlled trials shows that targeting lower arterial oxygen saturations does not decrease the likelihood of NEC; indeed, it increases the risk of NEC or death.  This information is most relevant to the paper, and it must be addressed in the Discussion section open (with 1 or more references), and also possibly in the Introduction.  Simply omitting this potentially disconfirming evidence is not appropriate.

Minor issues:

 To improve readability for researchers not intimately familiar with this topic, the authors should consider minimizing nonstandard and nonintuitive abbreviations.  For example, secretory components (SC) in line 60, 61, 70, etc., could be written without abbreviation.

Similarly, in my opinion, the leftmost column of Table 1 should contain the molecule names rather than abbreviations, so that readers do not have to keep referring to the bottom of a very long table in order to identify some of the less common molecules.  Abbreviations at the bottom of the table are still needed, since various molecules are cross-referenced in the right–most column.  Authors may wish to consider alphabetizing the abbreviations at the bottom of the table, to make them easier to find.

The figure illustrates relationships simply, but it probably should indicate the "entry points" for molecular oxygen, i.e., the mediators that oxygen generates or with which it directly interacts.  In addition, it is not clear why in the top/middle of the figure there is a pinkish NOS preceding NO, and a tan-colored NOS following NO.  How are these versions of NOS different?  If they are, the authors should indicate the differences in the legend and if necessary, clarify them in the body text.

I was unable to verify that reference 48 supports the statement in lines 187 and 188, which suggests that NEC risk is actually reduced.

In line 231, the authors state that in their “study”, they "systematically reviewed" mechanisms of hyperoxic injury to neonatal intestine.  This wording is misleading, since the manuscript does not report on a study as it is commonly understood, nor is it a systematic review.  A systematic review must include formal methodology with a protocol for searching, selecting and synthesizing the pertinent information.  This protocol should be designed and registered prior to initiating the systematic review.  The present manuscript contains what is more aptly described as a narrative review.

References need additional attention to detail.  For example, the format of the journal names should be uniform, with appropriate abbreviations per MDPI format.  Reference 48 is missing the journal name, and reference 53 is missing page numbers.

Author Response

Author Response to Reviewer 2 Comments:

The paper is generally well written, and it introduces the topic in a manner that is useful for both clinician scientists and bench researchers with an interest in hyperoxic injury and/or NEC. Below are some comments for the authors to consider, which may make the paper more accessible or useful to potential readers.

Major issues:

The major weakness of this paper is that while it addresses potential mechanisms of hyperoxic injury to the neonatal intestine that may contribute to NEC, it fails to address the clinical evidence on the association between intestinal oxygenation and intestinal injury.  Evidence from randomized, controlled trials shows that targeting lower arterial oxygen saturations does not decrease the likelihood of NEC; indeed, it increases the risk of NEC or death.  This information is most relevant to the paper, and it must be addressed in the Discussion section open (with 1 or more references), and also possibly in the Introduction.  Simply omitting this potentially disconfirming evidence is not appropriate.

Response:

Thanks for the reviewer’s comment. We agreed on the importance of targeting oxygen saturation influence in preterm care. We added “The optimal target range of oxygen saturation is 91%-95% in infants less than 28 weeks gestation [78]. There was increased mortality and NEC in the low oxygen saturation target group (85–89%) compared to the high SpO2 group (91–95%) [79].” in the “Future studies on hyperoxia-induced intestinal injuries “part (lines 249-252).

Minor issues:

To improve readability for researchers not intimately familiar with this topic, the authors should consider minimizing nonstandard and nonintuitive abbreviations. For example, secretory components (SC) in line 60, 61, 70, etc., could be written without abbreviation.

Response: Thanks for the reviewer’s suggestion. The abbreviations SC were deleted in the revised version.

Similarly, in my opinion, the leftmost column of Table 1 should contain the molecule names rather than abbreviations, so that readers do not have to keep referring to the bottom of a very long table in order to identify some of the less common molecules.  Abbreviations at the bottom of the table are still needed, since various molecules are cross-referenced in the right–most column.  Authors may wish to consider alphabetizing the abbreviations at the bottom of the table, to make them easier to find.

Response: Thanks for the reviewer’s comment. We have corrected Table 1 according to the reviewer’s suggestion.

The figure illustrates relationships simply, but it probably should indicate the "entry points" for molecular oxygen, i.e., the mediators that oxygen generates or with which it directly interacts. In addition, it is not clear why in the top/middle of the figure there is a pinkish NOS preceding NO, and a tan-colored NOS following NO.  How are these versions of NOS different?  If they are, the authors should indicate the differences in the legend and if necessary, clarify them in the body text.

Response: Thanks for the reviewer’s comment. We have amended Figure 1. We corrected the sentence to “Hyperoxia increases NOS and causes NO dysregulation and NO interacts with oxygen to became oxidated. Excessive NO induces the production of ROS. Besides, hyperoxia increases the production of ROS.” in Figure legend 1. Besides, we also revised 4.1. Nitric oxide and 4.3. Reactive oxygen species are as follow.

4.1. Nitric oxide

Nitric oxide (NO) is a free radical and can be a proinflammatory mediator as a signaling molecule to interact with oxygen to become oxidized, which can reaction is regulated by nitric oxide synthases (NOS), inhibits platelet adhesion, prevents mast cell activation, and functions as an antioxidant [35, 36]. High NO levels disrupt actin cytoskeletons, inhibit ATP formation, dilate cellular tight junction, and increase intestinal permeability [37]. Under hyperoxic conditions, NO dysregulation occurs and NOS II protein concentrations in the villus, crypts, submucosa, and muscularis are increased [17]. Excessive NO production leads to mucosal injury and may cause NEC [35, 38, 39].

4.3. Reactive oxygen species

Reactive oxygen species (ROS) are a chemically defined group of reactive molecules derived from molecular oxygen and ROS are produced by mitochondria. Low levels of ROS are related to cell proliferation and differentiation. Hyperoxia is associated with increased production of ROS, and excessive ROS induces apoptosis, cell autophagy, and DNA oxidative damage. A redox imbalance under hyperoxic conditions causes inflammation via the NF‑κB and TNF pathways [25, 36], which leads to an intestinal inflammation cascade and, ultimately, mucosal damage [29]. Mucosal damage induces the production of proinflammatory cytokines including IFN-g and IL-1. ROS also can modify cellular structure and function through covalent changes of NO and cause NO dysregulation [46]. Under conditions of hyperoxia, ROS promotes inflammatory cascades in the gut and may relate to inflammatory bowel disease [18, 47].

I was unable to verify that reference 48 supports the statement in lines 187 and 188, which suggests that NEC risk is actually reduced.

Response: Thanks for the reviewer’s comment. We verified that NEC risk is actually reduced in Reference 48.

In line 231, the authors state that in their “study”, they "systematically reviewed" mechanisms of hyperoxic injury to neonatal intestine.  This wording is misleading, since the manuscript does not report on a study as it is commonly understood, nor is it a systematic review.  A systematic review must include formal methodology with a protocol for searching, selecting and synthesizing the pertinent information.  This protocol should be designed and registered prior to initiating the systematic review.  The present manuscript contains what is more aptly described as a narrative review.

Response: Thanks for the reviewer’s comments. We rephrased the sentence to “In this study, we reviewed the mechanism underlying neonatal intestinal damage caused by hyperoxia at a histological and molecular level” and revised it according to the reviewer’s suggestions.

References need additional attention to detail.  For example, the format of the journal names should be uniform, with appropriate abbreviations per MDPI format.  Reference 48 is missing the journal name, and reference 53 is missing page numbers.

Response: We reformatted the References section and added Reference 48 and the page number of Reference 53.

Reviewer 3 Report

Comments to the Authors of manuscript number: ijms-2025892 entitled “Molecular Mechanisms of Hyperoxia-Induced Neonatal Intestinal Injury”.

I read carefully, but I cannot understand. The authors seem to want to explain the issue, but they wrote the topic, next a few sentences, and next point. where examples, where clinical cases, where explanation, description. It is rather in the form of overview of the most important events listed point by point. Moreover, it does not presents something new.

1. L 15 – histological

2. L16 – microbiota influence- is not clear.

3. L 18 – “redox” -? A complimentary process which involves electron transfer between reactants. Or it influences ROS? = Reactive Oxygen Species (ROS) is a phrase used to describe a number of reactive molecules and free radicals derived from molecular oxygen.

4. L 18-19 it is not understand at all

5. L20- redox balance is not suitable phrase

6. L 32 – reference should be added

7. L 37 – embryonic intestine development

8. L 44 - 45 this should be changed, the meaning after translation is different than that expected

9. L 46-47 The intestinal epithelium is part of the intestinal mucosa.

10. the part 2  is very chaotic. Why animals are involved in humans. Parts relating to humans and animals should be separated. It should be form the logical whole.

11.L 54 – it is not known if that maternal inflammation during pregnancy or in lactating, and when this development is altered? At weaning? This sentence is not clear

12. maybe the development in humans should be described with examples of clinical observation, and the development in animals with studies

13. L 59 – neonatal?

14. part 3 – what about other parts of intestinal tract?

15. L 74 – in whom? Animals or humans? At what age?

16. L 75 – lack or excess?

17. L 70 – it the same study?

18. L 82 – where and in whom? It is not clear

19. L 107 -  what is distal villus or proximal?

20. L 111-120 – what about it? It should be discussed on the basis of the intestine hyperoxia

21. part 5 - this part is in slogan style. It des not introduce nothing new.

22. L 237 – Authors wrote that “Supplemental oxygen therapy is essential for preterm neonates with respiratory disorders”, while in L 29 “High tissue oxygenation increases oxidative stress and leads to tissue injury”.

Author Response

Author Response to Reviewer 3 Comments:

I read carefully, but I cannot understand. The authors seem to want to explain the issue, but they wrote the topic, next a few sentences, and next point. where examples, where clinical cases, where explanation, description. It is rather in the form of overview of the most important events listed point by point. Moreover, it does not presents something new.

  1. L 15 – histological

Response: Thanks for the reviewer’s comment. We corrected ‘Histologic’ to ‘Histological’ on line 15.

  1. L16 – microbiota influence- is not clear.

Response: We corrected “Histologic changes include mucosal damage and vascular changes with microbiota influence” to “Histological changes include mucosal damage and vascular changes” on line 15 in the revised version.

  1. L 18 – “redox” -? A complimentary process which involves electron transfer between reactants. Or it influences ROS? = Reactive Oxygen Species (ROS) is a phrase used to describe a number of reactive molecules and free radicals derived from molecular oxygen.

Response: Thanks for the reviewer’s comment. We corrected the sentences to “Hyperoxia-induced intestinal injuries are influenced by several molecular factors, including the NF-κB and Nrf2 pathways, which are essential to maintain the balance of oxidative stress and antioxidants and prevent cell apoptosis and tissue inflammation” on lines 16-18.

  1. L 18-19 it is not understand at all

Response: We corrected the sentences “Intestinal inflammation may cause intestinal injury, necrotizing enterocolitis, and death” to “Intestinal inflammation can lead to intestinal damage and death of the intestinal tissue, such as necrotizing enterocolitis” on lines 18-20 in the revised version.

  1. L20- redox balance is not suitable phrase

Response: We corrected the sentence “Some antioxidant cytokines and molecules play a role in maintaining redox balance” to “Some antioxidant cytokines and molecules play a role in preventing cell apoptosis and tissue inflammation from oxidative stress” on lines 20-21 in the revised version.

  1. L 32 – reference should be added

Response: We corrected the sentence and added the references as “Previous studies in pediatric medicine have focused on the mechanisms underlying hyperoxia-induced lung injury [3-6]”.

  1. Torbati D, Tan GH, Smith S, Frazier KS, Gelvez J, Fakioglu H, Totapally BR: Multiple-organ effect of normobaric hyperoxia in neonatal rats. Journal of critical care 2006, 21(1):85-93; discussion 93-84.
  2. Giusto K, Wanczyk H, Jensen T, Finck C: Hyperoxia-induced bronchopulmonary dysplasia: better models for better therapies. Dis Model Mech 2021, 14(2).
  3. Zhu X, Lei X, Wang J, Dong W: Protective effects of resveratrol on hyperoxia-induced lung injury in neonatal rats by alleviating apoptosis and ROS production. The journal of maternal-fetal & neonatal medicine: the official journal of the European Association of Perinatal Medicine, the Federation of Asia and Oceania Perinatal Societies, the International Society of Perinatal Obstet 2020, 33(24):4150-4158.
  4. Bhandari V: Hyperoxia-derived lung damage in preterm infants. Semin Fetal Neonatal Med 2010, 15(4):223-229.

  1. L 37 – embryonic intestine development

Response: We corrected the subtitle from “Embryonic development of the intestine” to “2. Embryonic intestine development”.

  1. L 44 - 45 this should be changed, the meaning after translation is different than that expected

Response: We corrected “The intestinal epithelium, which covers the largest surface area of the body, is the major crucial barrier to the outside environment” to “The intestinal epithelium, which is the largest surface area of the body, is the major crucial barrier to the outside environment”.

  1. L 46-47 The intestinal epithelium is part of the intestinal mucosa.

Response: We corrected “The intestinal epithelium forms a barrier between the intestinal lumen and the intestinal mucosa” to “The intestinal epithelium forms a barrier between the intestinal lumen and the intestinal interstitium” on lies 47-48.

  1. the part 2 is very chaotic. Why animals are involved in humans. Parts relating to humans and animals should be separated. It should be form the logical whole.

Response: This part is to emphasize rodents are suitable animal models for studying O2–related acute intestinal damage. We divided the sentences into a new section: “Humans have long gestation periods; the gastrointestinal tract is mostly formed during gestation [10, 11]. In rodents with short gestation periods, such as rats and mice, the intestine is relatively immature at birth and becomes mature 2 weeks after birth [12]. These features increase the susceptibility of newborn rodents to hyperoxia. Rodents are suitable animal models for studying O2 toxicity–related acute intestinal damage” on lines 48-53.

11.L 54 – it is not known if that maternal inflammation during pregnancy or in lactating, and when this development is altered? At weaning? This sentence is not clear

Response: We corrected “In a mouse model, maternal inflammation altered the gastrointestinal tract of offspring [10]” to “In a mouse model, maternal inflammation during pregnancy altered the gastrointestinal tract of offspring [13].” on lines 55-56

  1. maybe the development in humans should be described with examples of clinical observation, and the development in animals with studies

Response: Thanks for the reviewer’s suggestion. We revise part 2 including adding the new section and pointing out the different discoveries in animal studies to avoid confusion. We divided the sentences into a new section: “Humans have long gestation periods; the gastrointestinal tract is mostly formed during gestation [10, 11]. In rodents with short gestation periods, such as rats and mice, the intestine is relatively immature at birth and becomes mature 2 weeks after birth [12]. These features increase the susceptibility of newborn rodents to hyperoxia. Rodents are suitable animal models for studying O2 toxicity–related acute intestinal damage” on lines 48-53.

  1. L 59 – neonatal?

Response: Thanks for the reviewer’s suggestion. We added ‘neonatal’ in “In neonatal rats reared in an O2-enriched environment, enterocytes were shorter and the surface of apical cells was flattened” on line 61.

  1. part 3 – what about other parts of intestinal tract?

Response:

Thanks for the reviewer’s advice. “Previous studies of hyperoxia-induced intestinal injury were focused on small intestines and large intestines”. We added this sentence to lines 57-59 in the revised version.

  1. L 74 – in whom? Animals or humans? At what age?

Response: We corrected the sentences to “Misalignment and distension of the basolateral intercellular space in the neonatal rat’s epithelium were also noted [16], and epithelial cell apoptosis and the mortality rate significantly increased [25, 27]” on lines 77-79.

  1. L 75 – lack or excess?

Response: We added ‘excess’ in “The small intestine in neonatal rats is sensitive to excess oxygen” on line 80.

  1. L 70 – it the same study?

Response: We deleted “Under conditions of hyperoxia, intestinal SIgA and IgA may change [27]. High level Under conditions of hyperoxia, SIgA and SC were markedly increased in neonatal rats [28]” and added “Moderate oxygen induced an increase and hyperoxia induced intestinal secretory components in neonatal rats and this might be brought to increase the intestinal SIgA. A large number of secretory components and SIgA may help in keeping optimal conditions from pathogen [22]” on lines 72-75 in the revised manuscript.

  1. L 82 – where and in whom? It is not clear

Response: We corrected “The number of Paneth cells also decreased” to “The number of Paneth cells also decreased in the neonatal rats” on lies 87-88.

  1. L 107 - what is distal villus or proximal?

Response: Thanks for the reviewer’s suggestion. We corrected “Under hyperoxic conditions, NO dysregulation occurs and NOS II protein concentrations in the distal villus, proximal villus, crypts, submucosa, and muscularis are increased” to “Under hyperoxic conditions, NO dysregulation occurs and NOS II protein concentrations in the crypts, submucosa, and muscularis are increased” on lines 101-102 in the revised version.

  1. L 111-120 – what about it? It should be discussed on the basis of the intestine hyperoxia

Response:

Thanks for the reviewer’s suggestion. We revised the statement “Under hyperoxic conditions, NF-κB transcription increases, which induces the production of proinflammatory mediators, such as tumor necrosis factor (TNF)-α and interferon (IFN)-γ [37]. NF-κB expression may be an important inflammatory mechanism of NEC [38] and may be related to inflammatory bowel disease [39].” to the “Under hyperoxic conditions, NF-κB transcription increases, which induces the production of proinflammatory mediators, such as tumor necrosis factor (TNF)-α and interferon (IFN)-γ [23, 42]. Excessive NF-κB expression may be an important inflammatory mechanism of NEC [43] and may be related to inflammatory bowel disease [37, 44]. NF-κB can directly repress Nrf2 signaling (the protective mechanism, it will be dis-cussed in part 5.1) to increase inflammation [45].” on lies 109-114. We hoped the revised version will let reader easy to understand the effect of excessive NF-κB expression under hyperoxic condition.

  1. part 5 - this part is in slogan style. It des not introduce nothing new.

Response:

Thanks for the reviewer’s comments. We do our best to review and organize the articles to let readers more easily to understand the Molecular Mechanisms of Hyperoxia-Induced Neonatal Intestinal Injury. We will keep investigative the protective mechanism under hyperoxic condition and the possible therapy from hyperoxia injury.

  1. L 237 – Authors wrote that “Supplemental oxygen therapy is essential for preterm neonates with respiratory disorders”, while in L 29 “High tissue oxygenation increases oxidative stress and leads to tissue injury”.

Response: Thanks for the reviewer’s suggestion. We rephrased the sentences to “Supplemental oxygen therapy is essential for preterm neonates with respiratory distress syndrome and other pulmonary conditions. The optimal target range of oxygen saturation is 91%-95% in infants less than 28 weeks gestation [78]. There was increased mortality and NEC in the low oxygen saturation target group (85–89%) compared to the high SpO2 group (91–95%) [79]” on lines 250-254.

Round 2

Reviewer 1 Report

The authors have made substantial revisions in response to the original critique. However the authors need resolve a problem.

  1. Reference 44 and 50 in line 168 were incorrect. Please add appropriate reference.

Response: Thanks for the reviewer’s comments. Correct references were added in “Nrf2 promotes IL-17D expression; hyperoxia promotes the nuclear translocation of Nrf2 and increases Nrf2 with IL-17D in intestinal epithelial cells [39, 44, 59].” on line 180.

Response to author Response: Reference 47 and 53 in revised manuscript don’t mention Nrf2 and IL-17D. Please delate them.

  1. Section 5.2 IL-10 describe just hypothesis. Please remove it.

Response: Thanks for the reviewer’s comment. We removed this section and remodified the figure in the revised manuscript.

Response to author Response: numbering following 5.1 section is incorrect. Please correct it.

Author Response

Author's Reply to the Review Report (Reviewer 1)

Comments and Suggestions for Authors

The authors have made substantial revisions in response to the original critique. However the authors need resolve a problem.

  1. Response to author Response: Reference 47 and 53 in revised manuscript don’t mention Nrf2 and IL-17D. Please delete them.

Response: Thanks for the reviewer’s comments. We had corrected the delete the reference 47 and 53 in Nrf2 and IL-17D section in line 191. We also corrected the part of table 1 in the revised version.

  1. Response to author Response: numbering following 5.1 section is incorrect. Please correct it.

Response: Thanks for the reviewer’s comments. The number of the sections following 5.1. section was corrected in the revised version.

Reviewer 3 Report

The review is written better than last time. I have no comments

Author Response

Author's Reply to the Review Report (Reviewer 3)

Comments and Suggestions for Authors

The review is written better than last time. I have no comments

Response: Thanks for the reviewer’s comments.